# The Effect of L-dopa on Postural Stability in Parkinson's Disease Patients

**Jacek Wilczyński** [1,*] and **Natalia Habik** [2]

[1] Department of Posturology, Hearing and Balance Rehabilitation, Faculty of Medicine and Health Sciences, Jan Kochanowski University, 25-317 Kielce, Poland

[2] Faculty of Medicine and Health Sciences, Jan Kochanowski University, 25-317 Kielce, Poland; habiknatalia@gmail.com

[*] Correspondence: wnoz_if@ujk.edu.pl; Tel.: +48-603-703-926

**Abstract:** The aim of the study was to evaluate the effects of L-dopa on postural stability in Parkinson's disease patients. In the study, we examined a group of 13 patients, members of the Parkinson's Association. The majority of subjects were women: 8 (61.538%), while 5 (38.462%) were men. These were patients with advanced, idiopathic Parkinson's disease. The study was performed at the Posturology Laboratory of the Faculty of Medicine and Health Sciences, UJK, Kielce (Poland). The duration of the illness was longer than 5 years. The daily L-dopa dose was between 600 and 1000 mg/d. Patients were tested for postural stability prior to taking the morning dose and again, 1 h after the 200-mg dose (Madopar 250 Tablets). The Biodex Balance System was applied in order to perform Postural Stability Testing. No statistically significant differences were found for the distribution of postural stability results before or after L-dopa administration. Nonetheless, it should be noted that all variables in the Postural Stability Test were slightly improved following L-dopa administration. The highest percentage (% Time in Zone) was noted in Zone A (the best), before (85.77%) and after L-dopa administration (95.23%). The highest % Time in Quadrant was in Quadrant IV (right posterior) both before (41.43%) and after L-dopa administration (49.54%). When comparing the distribution of postural stability variables before and after L-dopa administration, there were no significant differences between women and men.

**Keywords:** L-dopa; Parkinson's disease; Postural Stability Test

---

## 1. Introduction

The etiology of Parkinson's disease is still not determined. The essence of this disorder is to decrease the level of dopamine in the black matter [1,2]. Progressive and chronic atrophic changes in the Central Nervous System (CNS) that control motor function and degenerative changes in vision, hearing, proprioceptive systems as well as those related to balance, affect postural stability in Parkinson's patients. In Parkinson's disease patients, progressive failure of the neuromuscular system causes an increase in the thresholds of sensory system activity and uncontrolled muscle stimulation. The balance control system, which include various structures of the central nervous system, may be treated as a control system with three input locations (vestibular, proprioceptive and visual), determining spatial positioning of the body's center of gravity. The main task of the postural stability control system is to maintain an optimal distance between the stability limit and the body's center of gravity projection. If full symmetry of the body is assumed, the center of gravity projection should fall exactly in the center of the support surface [3,4]. Patients are prone to falls. They suddenly lose balance, ending in a fall (pulsion), especially with sudden head movements [5,6]. There may be a preference for backward (retropulsion), forward (propulsion) or lateral (lateropulsion) pulsion. Increases in the time of double

support and reduced walking speed occur [7,8]. The so-called tunnel effect consisting in taking small steps before a narrowing along a path of movement also takes place. Furthermore, the so-called freezing symptom may also appear, i.e., sudden obstruction while walking. Often, the patient cannot lift his/her foot up from the floor to take another step [9,10]. Sudden immobilization can cause loss of balance [11]. When the movement lock occurs during turning or turning around, it may cause a fall. Walking disorders are the most common cause of falls [12].

L-dopa (Lat. *Levodopum*) is a drug alleviating the symptoms of Parkinson's disease—a natural amino acid, a catecholamine produced in the process of tyrosine hydroxylation which is the resultant of the reaction led by tyrosine hydroxylase [13]. It is a dopamine precursor increasing this neurotransmitter's concentration in the brain. History has confirmed that L-dopa is a unique drug, and up to this day, no other invention has been created that would surpass its power of action [14]. However, in pharmacotherapy, the prototype is still being improved by creating its new form, for example: enteral, inhaled, long-acting L-dopa, etc. The introduction of L-dopa has caused many previously unknown concepts, such as fluctuations, dyskinesias and the spectrum of neuropsychiatric disorders to be discovered. It has definitely extended the survival time of patients [15].

The first stage of Parkinson's disease, lasting about 5 years, is mainly characterized by the presence of motor disorders, which are its axial symptoms (bradykinesia and resting tremor and/or muscle stiffness, and/or postural disturbances). In the later period, disabilities and so-called motor complications intensify in the form of shortening the proper response time to dopaminergic treatment (movement fluctuations) and movements in the form of dystonic dyskinesia (associated with lack of drug action) which are involuntary or chorea (associated with excessive drug activity). The therapeutic window of opportunity is narrowed and the time of good motor functioning is shortened. Fluctuations in movement and dyskinesia have complex mechanisms connected both with disturbed pharmacokinetics and pharmacodynamics. Disturbed pharmacokinetic mechanisms are mainly problems with the absorption of L-dopa and competition with amino acids during its passage from the intestines to the blood and across the blood-brain barrier. At a pharmacodynamic level, these are changes in the amount and sensitivity of dopamine receptors in the striatum. Thanks to the L-dopa stimulation strategy, it is possible to shorten the *off* time, extend the *on* states and reduce the severity of symptoms in the *off* period, which reduces the severity/duration of chorea dyskinesia and minimizes the symptoms of nocturnal disability [16].

Although L-dopa is highly effective, it is not ideal. It is associated with many problems such as fluctuations, dyskinesias, psychotic, autonomic or behavioral disorders [17]. Researchers are now learning how to combine it with other drugs such as dopamine agonists, or how it can be safely used after deep brain stimulation [18]. The main therapeutic goal in Parkinson's disease is to improve motor efficiency by optimally prolonging the *on* the period without the severity of chorea dyskinesia and non-cardiac symptoms. Posture- and gait-related disorders are one of the first changes signaling the disease. Therefore, physiotherapeutic procedures should firstly be aimed at improvement in this area. These symptoms predispose to the occurrence of falls, dangerous due to the risk of injuries and trauma, as a result of which the patient may be devoid of his/her ability to move. In relation to the-above, the aspect of the postural stability testing associated with L-dopa pharmacotherapy is of great importance. Early recognition of balance disturbances combined with targeted physiotherapy can reduce the negative effects and thus, improve the efficiency of patients. The study aim was to assess the effects L-dopa have on postural stability among patients with advanced idiopathic Parkinson's disease.

## 2. Materials and Methods

The study group comprised 13 patients who belonged to the Kielce, Poland Parkinson Disease Association. There were 8 females (61.538%) and 5 males (38.462%). The study took place in November 2013, in the Posturology Laboratory at the Institute of Physiotherapy by the Faculty of Medicine and Medical Sciences, Jan Kochanowski University in Kielce. The examined patients suffered from advanced, idiopathic Parkinson's disease. They still responded well to L-dopa. The duration of the

disease was more than 5 years. The daily dose of L-dopa stayed in between the range of 600–1000 mg/d. The patients were subjected to postural stability examination before the morning dose of the drug, and again, 1 h after taking 200 mg of L-dopa (Madopar 250 cap). All procedures performed in the case of studies with human participants took place in accordance with the ethical standards approved by the institutional and/or committee for national research as well as the Declaration of Helsinki from 1964, along with later amendments or analogous ethical standards. The patients were given information regarding the study aim. All of the patients expressed written consent to be participants in this study. The study was conducted in a non-invasive manner and was free of charge. The patients willingly took part in the experiment, perceiving it as a concern for their health. The Unified Parkinson's Disease Rating Scale, that is, the UPDRS motor test, was performed before testing on the Biodex Balance System Platform.

Postural stability was assessed with the use of the Biodex Balance System platform. Postural Stability Testing was performed with open eyes and with both of the patient's feet positioned on a stable surface. The platform was blocked, meaning that it was fully stable. After entering the patient's personal information and body height, their position was determined. For this reason, points of reference were considered to be the platform axes as well as the centre line of the foot. In order to determine the position, the angles of the feet positioning via the centre line (right and left foot separately) were entered into the system and displayed on the screen.

The Postural Stability Test comprised 3, 20-s trials which were divided by a 10-s interval. During the examination, the patient focused his/her eyesight on a monitor, where a characteristic dot (*COP—Centre of Pressure*) appeared. The patient was instructed to shift body balance in such a manner that the dot (COP) maintain in the middle of the circle appeared on the screen, which was at the point where the coordinate axes intersected. At the time of examination, it was allowed to verbally correct the patient allowed. All of the parameters registered by the posturological platform were collected in an absolutely non-invasive manner; the device could be safely applied in the case of the whole group.

Overall Stability Index (°) reflects variability of platform positioning with regarding to the horizontal plane. This is expressed in degrees and for all movements performed during the test. Its high value is evidence of the great number of movements performed at the time of the test. The Anterior-Posterior Stability Index (°) mirrors platform displacement variable in the case of sagittal plane movements, which again, are expressed in degrees. Medial-Lateral Stability Index (°) regards platform displacement variability concerning frontal plane movements in degrees. The score achieved by the patient in the Postural Stability Test depended on the how many times the patient swayed from the center, meaning that postural stability is better when the result is lower. The percentage of time in zone (%)—index regarding the time a patient spent in a given zone. Target zones A, B, C and D are the same, with respect to the platform tilt degree. These zones are determined by concentric circles, the mid part is in the platform center: Zone A: from 0 to 5 degree deviation with regard to the horizontal plane; Zone B: from 6 to 10 degree deviation with regard to the horizontal plane; Zone C: from 11 to 15 degree deviation with regard to the horizontal plane; Zone D: from 16 to 20 degree deviation with regard to the horizontal plane. Time in Quadrant (%)—this index concerns the time a patient spends in a given quadrant. Quadrants represent 4 quadrants of the graph used for testing between axes X and Y: Quadrant 1: the right anterior, Quadrant 2: the left anterior, Quadrant 3: the left posterior, Quadrant 4: the right posterior.

The score achieved by the patient in the Postural Stability Test depended on the number of times s/he swayed from the center, meaning that better postural stability is observed in the case of a lower result. The Biodex Balance System balance platform was used to assess postural stability. The Postural Stability Test was performed in a standing position on a stable surface with open eyes. The patient's position was determined after entering his/her personal information and body height. For this purpose, the central lines of the feet and the axes of the platform were used as reference points. In order to determine the position the positioning angle of the feet was entered using the center-line

on the screen (scale of 0°–45°, right and left foot separately, e.g., 25° in the case of the left foot and 30° for the right one) and heel position (scale used: B–J, 1–21, right and left foot separately, e.g., F7 for left foot and E15 for right one). The Postural Stability Test comprised 3, 20-s trials, which were separated by 10-s intervals. Before beginning the test, each subject was familiarized with all issues regarding the test. To ensure safety of the subjects, each session started with the platform being in a locked position. After being switched on or after 3 min of non-use, the Balance System automatically set the platform in a locked position. Also, before examination, the supports and monitor were adjusted to ensure the patient comfort and safety. During the test, due to safety measures, each examined subject was accompanied by a physical therapist during the tests by securing the patient in the event of a possible fall. The main task of the postural stability control system is to maintain optimum distance of the body's center of gravity from the stability limit. If full symmetry of the body is assumed, the center of gravity projection should fall exactly in the center of the support surface. Anatomical body asymmetries found in the sagittal and/or frontal planes, the distribution of sensory inputs and various biomechanical properties of the body in particular directions cause the center of gravity projection onto the supporting surface to not be in the center of this field.

Statistical analysis was conducted using the PQStat package, version 1.6. Variables presented according to gender were compared using the Mann–Whitney U test. Differences between results prior to and after administration of L-dopa were analysed with the use of the Wilcoxon Signed Ranks Test. Statistical significance was set at the level of $p < 0.05$.

## 3. Results

Comparing the distribution of postural stability variables before and after L-dopa administration showed no significant differences between women and men (Tables 1 and 2).

**Table 1.** Anthropometric variables.

| Anthropometric Variables | Sex | Arithmetic Mean | Standard Deviation | Minimum | Lower Quartile | Median | Upper Quartile | Maximum | Mann–Whitney U Test |
|---|---|---|---|---|---|---|---|---|---|
| Age | Total | 70.69 | 10.78 | 52.00 | 63.00 | 77.00 | 79.00 | 85.00 | Z = 0.5145 p = 0.6069 |
| | Women | 70.88 | 11.37 | 57.00 | 61.50 | 72.50 | 79.50 | 85.00 | |
| | Men | 70.40 | 11.06 | 52.00 | 68.00 | 77.00 | 77.00 | 78.00 | |
| Body height (cm) | Total | 165.2 | 8.80 | 151.0 | 158.0 | 166.0 | 175.0 | 176.0 | Z = 1.6384 p = 0.1013 |
| | Women | 161.7 | 8.48 | 151.0 | 156.5 | 158.0 | 167.5 | 175.0 | |
| | Men | 170.2 | 7.26 | 160.0 | 165.0 | 175.0 | 175.0 | 176.0 | |
| Body mass (kg) | Total | 68.76 | 11.95 | 52.10 | 59.60 | 68.80 | 76.70 | 86.70 | Z = 1.2443 p = 0.2134 |
| | Women | 64.73 | 10.58 | 52.10 | 58.08 | 64.35 | 69.63 | 84.20 | |
| | Men | 75.22 | 12.15 | 58.90 | 67.30 | 76.70 | 86.50 | 86.70 | |
| BMI | | 25.63 | 2.46 | 20.90 | 24.70 | 26.30 | 27.80 | 28.30 | 25.63 | Z = 0.3670 p = 0.7136 |
| | | 25.53 | 2.68 | 20.90 | 24.10 | 26.40 | 27.58 | 28.20 | 25.53 | |
| | | 25.80 | 2.35 | 23.00 | 24.70 | 24.80 | 28.20 | 28.30 | 25.80 | |
| Metabolic age (MA) | Total | 57.69 | 10.07 | 37.00 | 51.00 | 62.00 | 64.00 | 70.00 | Z = 0.4428 p = 0.6579 |
| | Women | 57.88 | 9.95 | 42.00 | 50.75 | 59.00 | 66.00 | 70.00 | |
| | Men | 57.40 | 11.44 | 37.00 | 62.00 | 62.00 | 62.00 | 64.00 | |

No statistically significant differences were found in postural stability between the pre- and post-L-dopa administration stages (Tables 2 and 3).

**Table 2.** Sex and postural stability before L-dopa administration.

| Variables of Postural Stability | Sex | Arithmetic Mean | Standard Deviation | Minimum | Lower Quartile | Median | Upper Quartile | Maximum | Mann–Whitney U Test |
|---|---|---|---|---|---|---|---|---|---|
| Overall Stability Index (°) | Total | 2.61 | 2.25 | 0.50 | 1.10 | 1.90 | 3.10 | 7.70 | Z = 0.7329 p = 0.4636 |
| | Women | 3.13 | 2.64 | 0.70 | 1.25 | 2.20 | 4.00 | 7.70 | |
| | Men | 1.78 | 1.28 | 0.50 | 0.90 | 1.20 | 3.10 | 3.20 | |
| Anterior-Posterior Stability Index (°) | Total | 1.99 | 2.30 | 0.34 | 0.72 | 1.20 | 1.40 | 7.60 | Z = 1.8298 p = 0.0673 |
| | Women | 2.75 | 2.70 | 0.50 | 1.18 | 1.35 | 3.48 | 7.60 | |
| | Men | 0.77 | 0.39 | 0.34 | 0.51 | 0.72 | 0.92 | 1.35 | |
| Medial-Lateral Stability Index (°) | Total | 1.15 | 1.02 | 0.20 | 0.40 | 0.70 | 1.50 | 3.00 | Z = 0.9568 p = 0.3387 |
| | Women | 0.95 | 0.86 | 0.20 | 0.30 | 0.75 | 1.20 | 2.70 | |
| | Men | 1.46 | 1.28 | 0.40 | 0.50 | 0.70 | 2.70 | 3.00 | |
| Zone A (%) | Total | 85.77 | 31.68 | 11.00 | 99.00 | 100.0 | 100.0 | 100.0 | Z = 0.0894 p = 0.9288 |
| | Women | 78.75 | 39.41 | 11.00 | 79.75 | 100.0 | 100.0 | 100.0 | |
| | Men | 97.00 | 6.16 | 86.00 | 99.00 | 100.0 | 100.0 | 100.0 | |
| Zone B (%) | Total | 13.38 | 29.60 | 0.00 | 0.00 | 0.00 | 1.00 | 81.00 | Z = 0.0894 p = 0.9288 |
| | Women | 19.88 | 36.81 | 0.00 | 0.00 | 0.00 | 19.50 | 81.00 | |
| | Men | 3.00 | 6.16 | 0.00 | 0.00 | 0.00 | 1.00 | 14.00 | |
| Zone C (%) | Total | 0.85 | 3.05 | 0.00 | 0.00 | 0.00 | 0.00 | 11.00 | Z = 0.6325 p = 0.5271 |
| | Women | 1.38 | 3.89 | 0.00 | 0.00 | 0.00 | 0.00 | 11.00 | |
| | Men | 0.00 | 0.00 | 0.00 | 0.00 | 0.00 | 0.00 | 0.00 | |
| Zone D (%) | Total | 0.00 | 0.00 | 0.00 | 0.00 | 0.00 | 0.00 | 0.00 | Z = 0 p = 0 |
| | Women | 0.00 | 0.00 | 0.00 | 0.00 | 0.00 | 0.00 | 0.00 | |
| | Men | 0.00 | 0.00 | 0.00 | 0.00 | 0.00 | 0.00 | 0.00 | |
| Quadrant I (%) | Total | 17.08 | 32.54 | 0.00 | 0.00 | 5.00 | 12.00 | 100.0 | Z = 1.0571 p = 0.2904 |
| | Women | 26.13 | 39.57 | 0.00 | 0.00 | 9.50 | 28.50 | 100.0 | |
| | Men | 2.60 | 3.21 | 0.00 | 0.00 | 1.00 | 5.00 | 7.00 | |
| Quadrant II (%) | Total | 9.00 | 15.47 | 0.00 | 0.00 | 0.00 | 7.00 | 46.00 | Z = 0.3344 p = 0.7381 |
| | Women | 12.88 | 18.89 | 0.00 | 0.00 | 0.00 | 25.25 | 46.00 | |
| | Men | 2.80 | 3.83 | 0.00 | 0.00 | 0.00 | 7.00 | 7.00 | |
| Quadrant III (%) | Total | 29.69 | 31.40 | 0.00 | 0.00 | 23.00 | 61.00 | 85.00 | Z = 0.8906 p = 0.3731 |
| | Women | 23.50 | 26.94 | 0.00 | 0.00 | 15.50 | 38.50 | 65.00 | |
| | Men | 39.60 | 38.58 | 0.00 | 4.00 | 37.00 | 72.00 | 85.00 | |
| Quadrant IV (%) | Total | 41.43 | 39.41 | 0.00 | 7.00 | 30.00 | 85.00 | 100.0 | Z = 0.9528 p = 0.3407 |
| | Women | 32.95 | 39.26 | 0.00 | 1.95 | 20.50 | 47.50 | 100.0 | |
| | Men | 55.00 | 39.84 | 7.00 | 28.00 | 49.00 | 95.00 | 96.00 | |

However, it should be noted that all parameters of postural stability slightly improved. After L-dopa administration, the Overall Stability Index decreased by 0.59 (°), while the Anterior-Posterior Stability Index (°) decreased by 0.33 (°), and the Medial-Lateral Stability Index (°) decreased by 0.33 (°). This indicates slightly better stability after administration of L-dopa (Table 4). Although no significant differences were noted in the % Time in Zone regarding particular A, B, C and D zones before or after L-dopa administration, it is worth noting that after drug administration, the percentages in Zone A (best) increased by 9.46 (%). The % Time in Zone B (slightly worse) after treatment with L-dopa decreased by 8.92 (%) and % Time in Zone C (worse) decreased by 0.54 (%) after administration. None of the patients were in Zone D (%) before or after administration (Table 4). The highest percentage of patients maintaining in Zone A (the best) was observed before (85.77%) and after L-dopa (95.23%) administration. There were also statistically significant differences in the % Time in Quadrant regarding individual Quadrants I, II, III, IV before and after L-dopa administration. However, it is important to note that after drug administration, the percentage of subjects in Quadrant I decreased by 2.62 (%). The % Time in Quadrant II (%) after L-dopa increased by 1.38 and the % Time in Quadrant III after treatment with L-dopa decreased by 4.07 (%). % Time in Quadrant IV after L-dopa administration increased by 8.11 (%) (Table 4). The highest percentage of time was spent in Quadrant IV (right-posterior), both before (41.43%) and after (49.54%) L-dopa administration (Table 4).

**Table 3.** Sex and postural stability after L-dopa administration.

| Variables of Postural Stability | Sex | Arithmetic Mean | Standard Deviation | Minimum | Lower Quartile | Median | Upper Quartile | Maximum | Mann–Whitney U Test |
|---|---|---|---|---|---|---|---|---|---|
| Overall Stability Index (°) | Total | 2.02 | 1.30 | 0.50 | 1.10 | 1.30 | 3.10 | 4.40 | Z = 0.7329 p = 0.4636 |
| | Women | 2.18 | 1.38 | 1.00 | 1.18 | 1.60 | 2.78 | 4.40 | |
| | Men | 1.78 | 1.28 | 0.50 | 0.90 | 1.20 | 3.10 | 3.20 | |
| Anterior-Posterior Stability Index (°) | Total | 1.66 | 1.30 | 0.40 | 0.60 | 1.20 | 2.70 | 4.20 | Z = 0.6605 p = 0.5089 |
| | Women | 1.79 | 1.38 | 0.50 | 0.68 | 1.35 | 2.50 | 4.20 | |
| | Men | 1.46 | 1.28 | 0.40 | 0.50 | 0.70 | 2.70 | 3.00 | |
| Medial-Lateral Stability Index (°) | Total | 0.82 | 0.58 | 0.10 | 0.60 | 0.70 | 0.90 | 2.20 | Z = 0.2239 p = 0.8228 |
| | Women | 0.80 | 0.63 | 0.10 | 0.53 | 0.70 | 0.90 | 2.20 | |
| | Men | 0.84 | 0.55 | 0.20 | 0.70 | 0.70 | 0.90 | 1.70 | |
| Zone A (%) | Total | 95.23 | 9.90 | 65.00 | 96.00 | 100.0 | 100.0 | 100.0 | Z = 0.3978 p = 0.6907 |
| | Women | 94.13 | 11.95 | 65.00 | 95.75 | 98.50 | 100.0 | 100.0 | |
| | Men | 97.00 | 6.16 | 86.00 | 99.00 | 100.0 | 100.0 | 100.0 | |
| Zone B (%) | Total | 4.46 | 8.89 | 0.00 | 0.00 | 0.00 | 4.00 | 31.00 | Z = 0.3978 p = 0.6907 |
| | Women | 5.38 | 10.56 | 0.00 | 0.00 | 1.50 | 4.25 | 31.00 | |
| | Men | 3.00 | 6.16 | 0.00 | 0.00 | 0.00 | 1.00 | 14.00 | |
| Zone C (%) | Total | 0.31 | 1.11 | 0.00 | 0.00 | 0.00 | 0.00 | 4.00 | Z = 0.6325 p = 0.5271 |
| | Women | 0.50 | 1.41 | 0.00 | 0.00 | 0.00 | 0.00 | 4.00 | |
| | Men | 0.00 | 0.00 | 0.00 | 0.00 | 0.00 | 0.00 | 0.00 | |
| Zone D (%) | Total | 0.00 | 0.00 | 0.00 | 0.00 | 0.00 | 0.00 | 0.00 | Z = 0 p = 0 |
| | Women | 0.00 | 0.00 | 0.00 | 0.00 | 0.00 | 0.00 | 0.00 | |
| | Men | 0.00 | 0.00 | 0.00 | 0.00 | 0.00 | 0.00 | 0.00 | |
| Quadrant I (%) | Total | 14.46 | 19.47 | 0.00 | 1.00 | 7.00 | 15.00 | 63.00 | Z = 1.9162 p = 0.0553 |
| | Women | 21.88 | 21.92 | 0.00 | 6.25 | 13.50 | 34.50 | 63.00 | |
| | Men | 2.60 | 3.21 | 0.00 | 0.00 | 1.00 | 5.00 | 7.00 | |
| Quadrant II (%) | Total | 10.38 | 18.64 | 0.00 | 0.00 | 5.00 | 8.00 | 68.00 | Z = 1.2818 p = 0.1999 |
| | Women | 15.13 | 22.82 | 0.00 | 0.75 | 6.50 | 19.25 | 68.00 | |
| | Men | 2.80 | 3.83 | 0.00 | 0.00 | 0.00 | 7.00 | 7.00 | |
| Quadrant III (%) | Total | 25.62 | 27.83 | 0.00 | 4.00 | 14.00 | 39.00 | 85.00 | Z = 0.5863 p = 0.5576 |
| | Women | 16.88 | 15.82 | 0.00 | 6.50 | 11.50 | 26.25 | 41.00 | |
| | Men | 39.60 | 38.58 | 0.00 | 4.00 | 37.00 | 72.00 | 85.00 | |
| Quadrant IV (%) | Total | 49.54 | 31.76 | 0.00 | 28.00 | 49.00 | 69.00 | 96.00 | Z = 0.2932 p = 0.7694 |
| | Women | 46.13 | 28.05 | 0.00 | 29.50 | 53.00 | 61.50 | 85.00 | |
| | Men | 55.00 | 39.84 | 7.00 | 28.00 | 49.00 | 95.00 | 96.00 | |

**Table 4.** Postural stability before and after L-dopa administration.

| Postural Stability Variables | Postural Stability Variables Before and After Drug Administration | Arithmetic Mean | Standard Deviation | Minimum | Lower Quartile | Median | Upper Quartile | Maximum | Wilcoxon Test |
|---|---|---|---|---|---|---|---|---|---|
| Overall Stability Index (°) | Before | 2.61 | 2.25 | 0.50 | 1.10 | 1.90 | 3.10 | 7.70 | Z = 0.8386 p = 0.4017 |
| | After | 2.02 | 1.30 | 0.50 | 1.10 | 1.30 | 3.10 | 4.40 | |
| Anterior-Posterior Stability Index (°) | Before | 1.99 | 2.30 | 0.34 | 0.72 | 1.20 | 1.40 | 7.60 | Z = 0.1177 p = 0.9063 |
| | After | 1.66 | 1.30 | 0.40 | 0.60 | 1.20 | 2.70 | 4.20 | |
| Medial-Lateral Stability Index (°) | Before | 1.15 | 1.02 | 0.20 | 0.40 | 0.70 | 1.50 | 3.00 | Z = 0.4318 p = 0.6659 |
| | After | 0.82 | 0.58 | 0.10 | 0.60 | 0.70 | 0.90 | 2.20 | |
| Zone A (%) | Before | 85.77 | 31.68 | 11.00 | 99.00 | 100.0 | 100.0 | 100.0 | Z = 0.2697 p = 0.7874 |
| | After | 95.23 | 9.90 | 65.00 | 96.00 | 100.0 | 100.0 | 100.0 | |
| Zone B (%) | Before | 13.38 | 29.60 | 0.00 | 0.00 | 0.00 | 1.00 | 81.00 | Z = 0.2697 p = 0.7874 |
| | After | 4.46 | 8.89 | 0.00 | 0.00 | 0.00 | 4.00 | 31.00 | |
| Zone C (%) | Before | 0.85 | 3.05 | 0.00 | 0.00 | 0.00 | 0.00 | 11.00 | Z = 0.0000 p = 1.0000 |
| | After | 0.31 | 1.11 | 0.00 | 0.00 | 0.00 | 0.00 | 4.00 | |
| Zone D (%) | Before | 0.00 | 0.00 | 0.00 | 0.00 | 0.00 | 0.00 | 0.00 | Z = 0.00 p = 0.00 |
| | After | 0.00 | 0.00 | 0.00 | 0.00 | 0.00 | 0.00 | 0.00 | |
| Quadrant I (%) | Before | 17.08 | 32.54 | 0.00 | 0.00 | 5.00 | 12.00 | 100.0 | Z = 0.3153 p = 0.7525 |
| | After | 14.46 | 19.47 | 0.00 | 1.00 | 7.00 | 15.00 | 63.00 | |
| Quadrant II (%) | Before | 9.00 | 15.47 | 0.00 | 0.00 | 0.00 | 7.00 | 46.00 | Z = 0.0000 p = 1.0000 |
| | After | 10.38 | 18.64 | 0.00 | 0.00 | 5.00 | 8.00 | 68.00 | |
| Quadrant III (%) | Before | 29.69 | 31.40 | 0.00 | 0.00 | 23.00 | 61.00 | 85.00 | Z = 0.4226 p = 0.6726 |
| | After | 25.62 | 27.83 | 0.00 | 4.00 | 14.00 | 39.00 | 85.00 | |
| Quadrant IV (%) | Before | 41.43 | 39.41 | 0.00 | 7.00 | 30.00 | 85.00 | 100.0 | Z = 0.9297 p = 0.3525 |
| | After | 49.54 | 31.76 | 0.00 | 28.00 | 49.00 | 69.00 | 96.00 | |

## 4. Discussion

Postural disorders are a fundamental problem in Parkinson's disease. They may be an expression of the disease but can also be an undesirable symptom of treatment. This has even been reflected in the diagnostic criteria of the disease. The specific response to drugs and the occurrence of fluctuations and dyskinesia associated with their use are considered typical Parkinson's disease symptoms. According to the European Federation of Neurological Societies/Movement Disorder Society—European Section (EFNS/MSD-ES), L-dopa is the most effective symptomatic drug in Parkinsonism treatment (strength of Recommendation: A) [19]. However, the occurrence of motor and postural complications (on average, in about 50% of patients after 3–5 years of treatment with this drug) hinder the functioning of a patient [20]. Posture- and motor-related disorders are accompanied by fluctuations in symptoms beyond motor sensory, autonomic and neuropsychiatric symptoms [21–23].

At the stage of advanced Parkinson's disease (PD), gait freezing is common but nonetheless, disabling. The relationship with dopaminergic medication may often be perceived as complex and often, even non-linear. That is why freezing may even occur when core Parkinsonian features (tremor, rigidity and/or bradykinesia) may seem to optimally be under control.

We conducted evaluation of the effect of Levodopa-carbidopa intrajejunal gel among a group of 7, non-demented patients with PD. They were characterized by prominent freezing episodes refractory to oral therapy adjustments. Clinical assessment was performed during the best *on* state prior to the application of Levodopa-carbidopa intrajejunal gel, while patients were subjected to standard oral Levodopa (O-LD) and treatment using infusion. The main outcome measures regarded changes in the Freezing of Gait (FOG) Questionnaire and UPDRS motor scores. In 4 of the seven studied 7case, the Levodopa-carbidopa intrajejunal gel dose was either equivalent or slightly higher, but for 3 patients, its level was lower in comparison to the O-LD dose noted at baseline. In the case of selected patients, Levodopa-carbidopa intrajejunal gel, compared to oral dopaminergic therapy, may improve freezing refractory [24].

The results of other studies suggest that static sway was noted as greater in IPD patients during the *off* state compared to subjects from the control group. Furthermore, it increased by L-dopa while it was reduced by GPI-DBS. During the dynamic task, L-dopa had greater impact than GPI-DBS when improving Start Time, but reduced spatial accuracy and task-related directional control. Combining the two therapies, it was noted that GPI-DBS prevented the increase in static sway induced by L-dopa while improving dynamic task accuracy. These findings show that GPI-DBS and L-dopa have different effects when considering the temporal and spatial aspects related to postural control in IPD. Furthermore, they demonstrate that GPI-DBS may counteract some adverse effects of L-dopa. Further studies among a larger population with GPI stimulators are needed in order to confirm the findings and to clarify the actual contribution of dyskinesia in impaired dynamic postural control [25].

The aim of a different study was to characterize control of postural stability control and the responsiveness of levodopa at the early stage of Parkinson's disease (PD). Postural sway was examined during quiet stance among 10 patients within 6 years of the onset of PD. This was done both before (OFF) and after (ON) the regular dosing of oral levodopa. Postural sway was registered with the use of a force platform during a 30 sec period, the patient keeping his/her eyes open, while3 dependent variables were assessed. Mild subclinical postural sway changes were noted at baseline for our patients. 5 out of 6 characteristics (mean postural sway, transversal sway, sagittal plane sway, intensity of sway and sway area) during the *on* condition showed it to be clearly beneficial. The mechanisms of postural control are affected at an early stage of PD and are modulated by dopamine [26].

The authors of other studies noted that PD patients show not only better functional status but also enhanced motor performance in the case of on-drug conditions. However, levodopa administration leads to increased postural sway. Postural sway occurred at baseline significantly more often in the case of on-drug patients than in the case of patients from the control group during tasks related to gaze-shifting. As it could be expected, acute administration of L-dopa did not cause an increase in eye, head, neck or lower back rotation among patients during gaze-shift tasks. Unexpectedly, levodopa

seemed to cause a significant increase in patients' postural control mechanisms (relative to the control group) during tasks regarding gaze-shifting. Nonetheless, it was not surprising that this adjustment was not large enough to allow patients to maintain postural sway to such an extent as the controls. Overall, levodopa administration appeared to destabilize patients—especially in the case of the lower back region [27].

LD or dopamine agonists may be used in patients requiring dopaminergic therapy. The most rational approach, except for patients at an advanced age, is initiation of treatment with dopamine agonists and its possible supplementation with LD, when monotherapy with dopamine agonists no longer produces effective effects. Dopamine agonists have been used in clinical practice for many years, but only the introduction of newer generations with lesser side effects (ropinirole, pramipexole, rotigotine) have widened the indications for their use in PD. The advantage of using dopamine agonists is a lower risk of developing dyskinesia and motor disturbances, whereas LD - greater efficacy, lower risk of hallucinations and excessive drowsiness, less risk of leg swelling, and avoidance of pulmonary fibrosis, retroperitoneal and pericardial effusions mainly caused by ergot alkaloids [19]. During early treatment of Parkinson's disease, very good control of motor symptoms can be obtained. There is an increased risk of drug complications, and at the advanced stage, it is no longer possible to obtain adequate control of movement disorders. When optimizing therapy, the response time to L-dopa should be considered—the so-called short- and long-duration response to levodopa as well as cessation of action after drug withdrawal. In the advanced form of the disease, the long-duration response (the consequence of the deterioration of the degenerative process) decreases, while the short-duration predominates, which can cause rapid changes in the patient's clinical condition [20]. L-dopa postoperative complications have varied clinical image. The mechanism of their formation is complex and has not yet been fully explained. Centered mechanisms (progression of nigrostriatal degeneration, changes in dopaminergic receptors, drug effects, effects of other neurotransmitters) and peripheral neuropathy (L-dopa absorption or metabolism) are suggested [19]. In Parkinson's disease, fluctuations in synaptic dopamine levels are preceded by the occurrence of motor fluctuations. It is known that greater predisposition to Parkinson's disease develops in men, and that the nature of the disease and the response to treatment depends on sex. Clinical image and the course of Parkinson's disease also differs among men and women. The bioavailability of L-dopa is higher in women [19]. However, our own research comparing the distribution of postural stability variables before and after L-dopa showed no significant differences between women and men [28].

## 5. Conclusions

1. No significant differences were noted between the distribution of postural stability results before or after L-dopa administration. However, it should be emphasized that all variables considered in the Postural Stability Test slightly improved after L-dopa administration.
2. The highest % Time in Zone regarded Zone A (the best), both before and after L-dopa administration.
3. The highest % Time in Quadrant regarded Quadrant IV (right-posterior) both before and after L-dopa administration.
4. Comparing the distribution of postural stability variables before and after administration of L-dopa, no significant differences between women and men were noted.

**Author Contributions:** J.W.: Data collection, preparation of methodology, research proper, statistical calculations, preparation of the manuscript, manuscript editing; N.H.: participation in patient examination, participation in manuscript editing.

**Funding:** This research received no external funding.

**Acknowledgments:** Authors wish to thank the recently deceased Paweł Półrola, who treated and cared for Parkinson's diseases patients.

**Conflicts of Interest:** The authors declare no conflict of interest regarding the publication of this paper.

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
