# Peer review of "The Effect of L-dopa on Postural Stability in Parkinson’s Disease Patients"

_applsci, doi:10.3390/app9030409_

Round 1
Reviewer 1 Report
The submitted work studied the effect of L-dopa on postural stability in Parkinson’s patients. The studying procedures were routine and the results were fully interpreted. Although no significant improvements had been found after L-dopa administration and comparison of the data between 8 women and 5 men showed no difference, those findings will still benefit the audience in in the field.
I believe this paper is of value to publish in applied sciences but needs revisions.
1. L-dopa effects have been widely studied. More background related to this study should be added.
2. I suggest the authors check the language and grammar in this paper. For example, redundant words should be removed from line 6 to line 10 on page 2.
Author Response
L-dopa effects have been widely studied. More background related to this study should be added:
The Biodex Balance System balance platform was used to assess postural stability. The Postural Stability Test was performed in a standing position on a stable surface with open eyes. After entering the patient’s personal data and body height, position was determined. For this purpose, the central lines of the feet and the axes of the platform were used as reference points. The position was determined by entering the positioning angle of the feet using the centre-line on the screen (scale 0°-45° separately for the right and left foot, e.g. 25° for the left foot and 30° for the right one) and heel position (scale B - J, 1-21, separately for the right and left foot, e.g. F7 left foot and E15 right one). The Postural Stability Test consisted of 3, 20-second trials, separated by 10-second intervals. Before beginning the test, each subject was familiarised with all issues regarding the test. To ensure the subject’s safety, each session began with the platform in a locked position. The Balance System automatically set the platform in a locked position after the being switched on or after a period of 3 minutes of its non-use. Also, before examination, the supports and monitor were adjusted to ensure the patient comfort and safety. During the test, due to safety measures, each examined subject was accompanied by a physical therapist during the tests by securing the patient in the event of a possible fall.
Reviewer 2 Report
The introduction should include more about the postural effects of levodopa and why you are doing the study. If you did do a UPDRS motor score before the testing this should be included. Even with a night without levodopa some patients are no really off and it would be good to show that patients were really off and on when you did the off and on testing. The methods and results are presented clearly. The conclusion again I think should focus a bit more on balance. There is a lot about freezing in both the intro and discussion but this is not something that you focused on at all during the study.
Round 2
Reviewer 1 Report
The manuscript has been revised extensively by the authors. Required background information about this study was added. Language and grammars were checked and revised. I believe it is appropriate to publish after those revisions.